# Accuracy of the electronic health record's problem list in describing multimorbidity in patients with heart failure in the emergency department

**Brandon L. King**[1]☯, **Michelle L. Meyer**[1‡], **Srihari V. Chari**[1‡], **Karen Hurka-Richardson**[1‡], **Thomas Bohrmann**[2‡], **Patricia P. Chang**[3‡], **Jo Ellen Rodgers**[4‡], **Jan Busby-Whitehead**[5‡], **Martin F. Casey**[1]☯*

1 Department of Emergency Medicine, University of North Carolina School of Medicine, Chapel Hill, North Carolina, United States of America, 2 Analytical Partners Consulting LLC, Raleigh, North Carolina, United States of America, 3 Division of Cardiology, Department of Medicine, University of North Carolina School of Medicine, Chapel Hill, North Carolina, United States of America, 4 Division of Pharmacotherapy and Experimental Therapeutics, University of North Carolina Eshelman School of Pharmacy, Chapel Hill, North Carolina, United States of America, 5 Division of Geriatric Medicine and Center of Aging and Health, University of North Carolina School of Medicine, Chapel Hill, North Carolina, United States of America

☯ These authors contributed equally to this work.
‡ MLM, SVC, KHR, TB, PPC, JER and JBW also contributed equally to this work.
* martin_casey@unc.edu

**Data Availability Statement:** All relevant data are within the paper and its Supporting information files.

## Abstract

Patients with heart failure (HF) often suffer from multimorbidity. Rapid assessment of multimorbidity is important for minimizing the risk of harmful drug-disease and drug-drug interactions. We assessed the accuracy of using the electronic health record (EHR) problem list to identify comorbid conditions among patients with chronic HF in the emergency department (ED). A retrospective chart review study was performed on a random sample of 200 patients age ≥65 years with a diagnosis of HF presenting to an academic ED in 2019. We assessed participant chronic conditions using: (1) structured chart review (gold standard) and (2) an EHR-based algorithm using the problem list. Chronic conditions were classified into 37 disease domains using the Agency for Healthcare Research Quality's Elixhauser Comorbidity Software. For each disease domain, we report the sensitivity, specificity, positive predictive value, and negative predictive of using an EHR-based algorithm. We calculated the intra-class correlation coefficient (ICC) to assess overall agreement on Elixhauser domain count between chart review and problem list. Patients with HF had a mean of 5.4 chronic conditions (SD 2.1) in the chart review and a mean of 4.1 chronic conditions (SD 2.1) in the EHR-based problem list. The five most prevalent domains were uncomplicated hypertension (90%), obesity (42%), chronic pulmonary disease (38%), deficiency anemias (33%), and diabetes with chronic complications (30.5%). The positive predictive value and negative predictive value of using the EHR-based problem list was greater than 90% for 24/37 and 32/37 disease domains, respectively. The EHR-based problem list correctly identified 3.7 domains per patient and misclassified 2.0 domains per patient. Overall, the ICC in comparing Elixhauser domain count was 0.77

**Funding:** BK received a grant from the ISTEM NHLBI T35 supported by grant number T35-HL134624 from the National Institutes of Health. URL: https://www.niddk.nih.gov/research-funding/process/apply/funding-mechanisms/t35 No - The funders had no role in study design, data collection and analysis, decision to publish, or preparation of the manuscript.

**Competing interests:** The authors have declared that no competing interests exist.

(95% CI: 0.71-0.82). The EHR-based problem list captures multimorbidity with moderate-to-good accuracy in patient with HF in the ED.

## Introduction

Patients with heart failure (HF) often suffer from a heavy burden of multimorbidity (i.e., the co-occurrence of two or more chronic medical conditions). On average, patients with HF have 4 to 8 chronic medical conditions in addition to their HF [1–3]. Worsening burden of multimorbidity is predictive of increased healthcare costs, polypharmacy, and risk of both hospitalization and death among patients with HF [1, 4, 5]. Potential mechanisms by which multimorbidity worsens outcomes include poor care coordination, polypharmacy and drug interactions, and conflicting medical recommendations across disease-specific guidelines [6–9]. Common chronic conditions that are recognized as complicating the care of patients with HF include anemia, sleep disordered breathing, respiratory disease, renal impairment, type 2 diabetes, thyroid disease, musculoskeletal disorders, arrhythmias, depression, and cognitive impairment [2, 10]. Despite the emerging importance of multimorbidity, there is a lack of systematic approaches to measure co-occurring chronic conditions, particularly in the setting of emergency care [11–13]. In clinical practice, physicians use the shared mental model of the 'problem list', a written list of medical problems requiring management, to rapidly communicate the overall burden of medical illness in a patient (Fig 1). The problem list is particularly important for emergency department (ED) physicians who routinely care for unfamiliar and undifferentiated patients. Given its importance in clinical practice, the problem list has been widely implemented in various types of electronic health records (EHRs) [14, 15]. The EHR-based problem list provides structured data that can be readily extracted in the pursuit of studying multimorbidity. Unfortunately, problem lists are not traditionally used for research due to the perception that they are mostly managed by primary care physicians and often incomplete or inaccurate [14, 16, 17]. However, advances in health informatics—including problem-based charting, clinical decision support, and linkage to payment mechanisms in the her [18–20]—as well as federal mandates for providers to demonstrate meaningful use of health information technology—have improved documentation of conditions within problem lists [21, 22]. In the context of these advances, EHR-based problem lists may provide an accurate picture of overall disease burden in multimorbid patients.

We hypothesized that problem lists may be reasonably accurate among ED patients with a previously documented diagnosis of chronic HF for two reasons: (1) temporal trends in

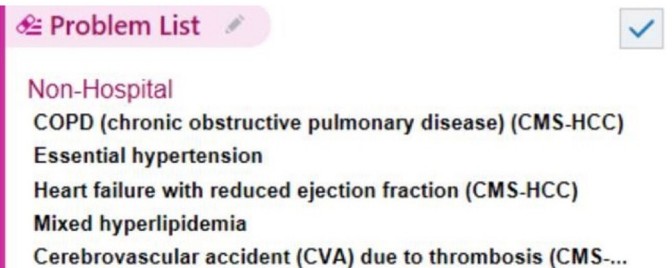

**Fig 1. Depiction of the electronic health record-based problem list.** Used with permission of 2021 Epic Systems Corporation.

increased use of health information technology and (2) patients with HF often have routine outpatient encounters in which EHR-based problem lists may be updated. Measurement of multimorbidity from the problem list requires the application of a categorization system. We sought to assess the accuracy of applying the Elixhauser comorbidity index, a widely studied clinical classification system traditionally applied to administrative and hospital billing data [23], that collapses thousands of diagnoses into 38 meaningful co-morbid domains.

## Materials and methods

### Study design

We performed a retrospective chart review study on a random sample of 200 patients with a known diagnosis of HF who presented to an ED. All patient encounters occurred at a southeastern academic medical center between January 1, 2019 and December 31, 2019. Data was obtained from the Carolina Data Warehouse for Health (CDW-H), a central data repository sourced from two electronic health records within the University of North Carolina Health Care System (UNC-HCS). The study was approved and granted a waiver of consent by the UNC Institutional Review Board (IRB #21–0002). Study reporting followed the 2015 STARD Guideline for Reporting Diagnostic Accuracy Studies [24].

### Participants

Patients were eligible for inclusion based on a documented diagnosis of HF on the EHR-based problem list. A comprehensive list of International Classification of Diseases, Tenth Revision, Clinical Modification (ICD-10-CM) diagnostic codes to identify HF were based on the Agency for Healthcare Research and Quality's (AHRQ) Elixhauser Comorbidity Software v2021.1 [25]. These codes included: I50x, I0981, I110, I130, I132, I5181, I97130, I97131, O2912x, R570, Z95811, Z97811, Z95812. Similar codes have had high specificity (98.8%), though moderate sensitivity (78.7%), in identifying patients with HF in administrative datasets [26]. More importantly, the application of the I50x diagnostic codes to EHR-based problem lists has been demonstrated to identify patients with HF with a high positive predictive value (PPV) of 96% [27]. Given that multimorbidity is predominantly a problem in older adults, we restricted our study population to patients ≥65 years old. Finally, only a patient's index ED visit in 2019 was eligible for random selection into the study cohort (i.e., repeat ED visits were excluded). Random selection was performed, using a random number sequence generator by the R Core Team [28]. To assess the quality of the random selection, patient characteristics of all eligible patients were compared to the characteristics of patients selected for chart review.

### Measuring multimorbidity

Although there is no consensus on the optimal approach to measure multimorbidity [29, 30], the Elixhauser comorbidity index has been validated for use with both ICD-9 and ICD-10-CM diagnostic codes [23, 31]. Though the Elixhauser index was originally intended to prognosticate future mortality based on 29 disease domains, the recent release of the AHRQ's updated Elixhauser comorbidity software for ICD-10-CM diagnostic codes (v2021.1) has expanded the number of disease domains to 38, thus providing a more comprehensive description of disease burden in patients [25]. As the AHRQ Elixhauser software is routinely updated and provided at no cost to researchers, the proposed methods provide an optimal clinical classification system to apply in the study of multimorbidity.

Chart review was performed as a gold standard measurement in capturing multimorbidity at the time of the ED encounter. A research assistant (BK) was trained in structured chart

review to extract data on chronic conditions from five areas in the EHR including: (1) ED provider documentation, (2) admission summary if available, (3) outpatient encounters and discharge summaries prior to the ED encounter of interest, and (4) 'Care Everywhere', which is a tool for health information exchange between healthcare systems. Height and weight were directly reviewed from EHR flowsheets to determine the presence of obesity. All documented conditions were mapped back to ICD-10-CM diagnostics codes and categorized into the 38 Elixhauser domains using the AHRQ's reference files [25]. All charts with unclear documentation or conflicting documentation were reviewed by an ED physician (MFC) for final adjudication. Except for echocardiogram reports, labs and radiographic data were not directly reviewed unless unclearly documented diagnoses required further adjudication. To assess the reliability of our gold standard measurement, 15% of the study sample was reviewed by a nurse practitioner (KHR) who was similarly trained in structured chart abstraction. Both chart reviewers were unblinded to the EHR-based problem list as most provider notes used pre-populated templates that extract data directly from the problem list. /par In parallel, multimorbidity was measured in the problem list by extracting ICD-10-CM codes tied to chronic conditions on the EHR-based problem list. Codes had to have a designation of active on the non-hospital problem list (i.e., presumably requiring outpatient management) at the time of the ED encounter. All active ICD-10-CM diagnostic codes were then entered into the Elixhauser comorbidity software for categorization into the 38 disease domains (including HF). The data analyst (TB) and research assistant (SVC) who measured multimorbidity from the problem list were blinded to the process and results of the chart review.

Finally, the AHRQ Elixhauser software calculates two summary indices, based on a patient's multimorbidity, to prognosticate a patient's risk of in-hospital mortality and readmission [32]. The weights were extracted from the software and used to calculate each index based on the data abstracted from the chart review. Similarly, the AHRQ software automatically calculated the indices based on the ICD-10-CM codes provided from the EHR-based problem list.

## Analysis

Descriptive statistics were used to characterize prevalence of each domain and distribution of disease burden as measured by chart review. To assess the reliability of the chart review, the prevalence-adjusted bias-adjusted kappa (PABAK) was calculated for each domain across the two chart reviewers (BK, KHR). In comparing the accuracy of the EHR-based problem list against chart review, we report the sensitivity, specificity, PPV, and negative predictive value (NPV) for each Elixhauser domain. Overall agreement between chart review and the EHR-based problem list was assessed by measuring the intraclass correlation coefficient (ICC) on three summary measures of multimorbidity including: (1) Elixhauser domain count, (2) Elixhauser readmission index, and (3) Elixhauser in-hospital mortality index. To obtain the ICC, we used a two-way, mixed effects models to obtain ICCs on consistency of agreement. We also report ICCs on absolute agreement as they provide complementary information [33]. Thresholds for reporting PABAK and ICC (values less than 0.5, between 0.5 and 0.75, between 0.75 and 0.9, and greater than 0.90 represent poor, moderate, good, and excellent agreement, respectively) were based on established guidelines [34, 35]. Bland-Altman analysis was performed to further characterize the degree of agreement between the two approaches in measuring Elixhauser domain count [36]. Finally, diagnostic accuracy by threshold classification was performed using receiver operating characteristic (ROC) curve analysis. A gold standard binary classification of Elixhauser domain count was created using the presence of multimorbidity (i.e., presence of 2 or more additional chronic conditions in accord with prior studies) as observed in the gold standard chart review [11]. Similar, Elixhauser readmission and

mortality indices were dichotomized into high or low scores based on the median value observed in the study population. Area under the curve (AUC) is reported as the measure of accuracy of the EHR-based problem list in characterizing the Elixhauser domain count, readmission index, and mortality index in accord with the binary classifications discussed above. All statistical analyses were performed in Stata MP/16.1 and SAS 9.4.

## Results

### Participants

A total of 66,600 ED encounters between January 1, 2019-December 31, 2019 were assessed for study eligibility. After identifying index visits by older patients (ie, ≥65 years old) with HF, there were 1,130 ED encounters eligible for random selection for further chart review (S1 Fig). In the study cohort, the mean age was 78.3 years (standard deviation [SD] 9.2) and 52.5% patients were female (Table 1). Most patients were white (65%), non-Hispanic (95.5%), and used Medicare as their primary payor (95.5%). The most prevalent HF phenotype observed in the study cohort was HF with preserved ejection fraction (60%). Patient characteristics in the study cohort were similar to those found in 1,130 ED encounters eligible for random selection (S1 Table).

**Table 1. Demographics of study sample and level of multimorbidity by subgroup.**

| | | N (%) | Mean number of Elixhauser domains (SD) | Minimum-maximum number of domains |
|---|---|---|---|---|
| Total | | 200 (100%) | 5.4 (2.1) | 1–10 |
| Age | | | | |
| | 65–69 | 48 (24%) | 5.6 (2.1) | 2–9 |
| | 70–74 | 27 (13.5%) | 6.2 (2.0) | 3–9 |
| | 75–79 | 35 (17.5%) | 5.5 (2.1) | 1–9 |
| | 80–84 | 32 (16%) | 5.1 (2.1) | 2–10 |
| | 85+ | 58 (29%) | 5.1 (2.1) | 1–10 |
| Gender | | | | |
| | Female | 105 (52.5%) | 5.6 (2.1) | 1–10 |
| | Male | 95 (47.5%) | 5.3 (2.1) | 1–10 |
| Race | | | | |
| | White | 130 (65%) | 5.6 (2.1) | 1–10 |
| | Black | 57 (28.5%) | 5.4 (2.0) | 2–10 |
| | Other | 13 (6.5%) | 4.5 (1.9) | 2–8 |
| Ethnicity | | | | |
| | Non-Hispanic | 191 (95.5%) | 5.5 (2.1) | 1–10 |
| | Hispanic | 9 (4.5%) | 4.8 (1.4) | 3–8 |
| Primary Payer | | | | |
| | Medicare | 191 (95.5%) | 5.4 (2.1) | 1–10 |
| | Non-Medicare | 9 (4.5%) | 5 (2.1) | 2–8 |
| HF Phenotype | | | | |
| | Reduced EF | 65 (32.5%) | 5.4 (2.0) | 1–9 |
| | Preserved EF | 120 (60%) | 5.6 (2.2) | 1–10 |
| | Unspecified | 15 (7.5%) | 4.7 (1.9) | 2–9 |

SD = standard deviation; HF = heart failure; EF = ejection fraction

## Multimorbidity by chart review (Gold standard)

There was a mean number of 5.4 Elixhauser domains identified in each patient with HF (SD 2.1; range: minimum 1 to maximum 10; Table 1). The mean Elixhauser readmission and mortality index scores were 18.0 (SD 7.5) and 19.0 (SD 13.2), respectively. The distribution of multimorbidity followed similar patterns when stratified by patient demographics including age group, race, ethnicity, and primary insurer. The top 10 most prevalent domains included: uncomplicated hypertension (90%), obesity (42%), chronic pulmonary disease (38%), deficiency anemia (33%), diabetes with chronic complications (30.5%), cerebrovascular disease (28%), depression (27%), valvular disease (26%), moderate renal failure (25.5%), and peripheral vascular disease (24%; Table 2). Among 37 possible domains, interrater reliability of structured chart review across reviewers was found to be moderate, good, and near perfect in 5, 8, and 18 domains, respectively (PABAK ranged between 0.67 to 1.0; S2 Table). PABAK was indeterminate for 6 domains as perfect alignment in either positive or negative cases results in a denominator of 0 in the calculation of kappa.

## Accuracy of EHR problem list

Extraction of multimorbidity data using ICD-10-CM diagnostics codes from the problem list identified a mean of 4.1 Elixhauser domains per patient with HF (SD 2.1; range: minimum 0 to maximum 11). The mean Elixhauser readmission and in-hospital mortality index scores of 16.4 (SD 6.7) and 19.5 (SD 11.7), respectively.

**Table 2. Diagnostic accuracy of measuring Elixhauser comorbid domains from the EHR-based problem list for the ten most prevalent disease domains.**

| Elixhauser domain 1 | Prevalence by chart review (gold standard) | Sensitivity | Specificity | PPV | NPV |
|---|---|---|---|---|---|
|  | % (N) | % | % | % | % |
|  |  | (95% CI) | (95% CI) | (95% CI) | (95% CI) |
| Cerebrovascular disease | 28% (56) | 57.10% | 99.30% | 97.00% | 85.60% |
|  |  | (43.2%-70.3%) | (96.2%-100.0%) | (84.2%-99.9%) | (79.4%-90.6%) |
| Chronic pulm. disease | 38% (76) | 76.30% | 99.20% | 98.30% | 87.20% |
|  |  | (65.2%-85.3%) | (95.6%-100.0%) | (90.9%-100.0%) | (80.6%-92.3%) |
| Deficiency anemias | 33% (66) | 69.70% | 97.80% | 93.90% | 86.80% |
|  |  | (57.1%-80.4%) | (93.6%-99.5%) | (83.1%-98.7%) | (80.3%-91.7%) |
| Depression | 27% (54) | 51.90% | 98.60% | 93.30% | 84.70% |
|  |  | (37.8%-65.7%) | (95.1%-99.8%) | (77.9%-99.2%) | (78.4%-89.8%) |
| Diabetes with chronic comp. | 30.5% (61) | 85.20% | 97.80% | 94.50% | 93.80% |
|  |  | (73.8%-93.0%) | (93.8%-99.6%) | (84.9%-98.9%) | (88.5%-97.1%) |
| Moderate renal failure | 25.5% (51) | 68.60% | 98.70% | 94.60% | 90.20% |
|  |  | (54.1%-80.9%) | (95.2%-99.8%) | (81.8%-99.3%) | (84.5%-94.3%) |
| Obesity | 42% (84) | 28.60% | 97.40% | 88.90% | 65.30% |
|  |  | (19.2%-39.5%) | (92.6%-99.5%) | (70.8%-97.6%) | (57.7%-72.4%) |
| Peripheral vascular disease | 24% (48) | 77.10% | 96.70% | 88.10% | 93.00% |
|  |  | (62.7%-88.0%) | (92.5%-98.9%) | (74.4%-96.0%) | (87.9%-96.5%) |
| Uncomplicated hypertension | 90% (180) | 77.80% | 100.00% | 100.00% | 33.30% |
|  |  | (71.0%-83.6%) | (83.2%-100.0%) | (97.4%-100.0%) | (21.7%-46.7%) |
| Valvular disease | 26% (52) | 80.80% | 98.60% | 95.50% | 93.60% |
|  |  | (67.5%-90.4%) | (95.2%-99.8%) | (84.5%-99.4%) | (88.5%-96.9%) |

EHR = electronic health record; PPV = positive predictive value; NPV = negative predictive value

The diagnostic accuracy of the EHR problem list as compared to chart review—as measured by sensitivity, specificity, PPV, and NPV—for the ten most prevalent domains is presented in Table 2. Among the top ten most prevalent domains, sensitivity ranged widely from 28.6% to 80.8%. Most conditions were above 75% sensitive with the exceptions of cerebrovascular disease (57.1%), deficiency anemia (69.7%), depression (51%), and obesity (28.6%). Conversely, reported specificities occurred within a narrow range of 96.7% to 100%. Correspondingly, the PPV of the ten most prevalent conditions ranged from 88.1% to 100%. The NPV also varied widely ranging between 33.3% to 93.8%. However, most domains had a NPV above 85% with the exceptions of uncomplicated hypertension (33.3%), obesity (65.3%), and depression (84.7%).

Similar variability (or lack thereof) was seen in the sensitivities and specificities among lower prevalence disease domains (S3 Table). In total, 13 domains had a PPV lower than 90% including: obesity (88.9%), peripheral vascular disease (88.1%), mild liver disease (87.5%), seizure disorder (85.7%), diabetes without chronic complications (81.8%), peptic ulcer disease (75%), paralysis (75%), other neurologic disorders (72.7%), weight loss (69.2%), complicated hypertension (54.5%), psychoses (33.3%), blood loss anemia (28.6%), and cancer-solid tumor without metastasis (28.6%). None of the lower prevalence domains had a NPV below 90%.

Perfect alignment between chart review and the problem list was observed in 35 patients (17.5%; Table 3). The EHR-based problem list correctly identified 3.7 Elixhauser domains per patient. Conversely, there was an average of 2 misclassified (i.e., either false positive or false negative) domains per patient. The average number of misclassified domains was observed to increase among patients with higher multimorbidity (i.e., higher domain count as reported by chart review). In measuring the agreement on Elixhauser domain count between chart review and EHR-problem list, we observed a consistency ICC of 0.77 (Table 4). Similar consistency ICCs were reported in the comparison of Elixhauser readmission (ICC 0.81) and in-hospital

**Table 3. Overall accuracy of the EHR-based problem list stratified by level of multimorbidity reported in chart review.**

| Elixhauser domain count by chart review (gold standard) | | | | | | | | | | | |
|---|---|---|---|---|---|---|---|---|---|---|---|
| | 1 | 2 | 3 | 4 | 5 | 6 | 7 | 8 | 9 | 10 | Total |
| Patients (N) | 2 | 13 | 25 | 33 | 31 | 32 | 24 | 24 | 14 | 2 | 200 |
| Proportion of patients with all domains captured by HER problem list (%) | 100% | 62% | 24% | 24% | 19% | 13% | 13% | 4% | 7% | 0% | 20% |
| Proportion of patients with perfect agreement between chart review and EHR problem list (%) | 100% | 54% | 24% | 24% | 13% | 13% | 13% | 0% | 7% | 0% | 18% |
| Average number of correctly classified domains by EHR-based problem list (%) | 1 | 1.5 | 1.8 | 2.4 | 3.5 | 4.4 | 5.2 | 5.1 | 6.4 | 8.5 | 3.7 |
| Average number of missed domains by EHR-based problem list* | 0 | 0.5 | 1.2 | 1.6 | 1.5 | 1.6 | 1.8 | 2.9 | 2.6 | 1.5 | 1.7 |
| Average number of misclassified domains by EHR-based problem list* | 0 | 0.6 | 1.6 | 1.8 | 2 | 1.9 | 2.1 | 3.3 | 2.9 | 2 | 2 |

*Rate of misclassified domains = false negative rate + false positive rate; Rate of missed domains = false negative rate alone; EHR = Electronic Health Record System

**Table 4. Comparison Elixhauser multimorbidity measures by chart review and EHR-based problem list.**

| | Multimorbidity meausrement | | Comparison of measurements | | |
|---|---|---|---|---|---|
| | Chart review (gold standard) | EHR-based problem list | Mean difference | Consistency ICC | Absolute ICC |
| | Mean (SD) | Mean (SD) | | ICC (95% CI) | ICC (95%CI) |
| Elixhauser domain count | 5.4 (2.1) | 4.1 (2.1) | 1.3 | 0.77 (0.71–0.82) | 0.62 (0.09–0.82) |
| Elixhauser readmission rate | 18.0 (7.5) | 16.4 (6.7) | 1.6 | 0.81 (0.75–0.85) | 0.79 (0.70–0.85) |
| Elixhauser in-hospital mortality index | 19.0 (13.2) | 19.5 (11.7) | -0.5 | 0.76 (0.69–0.81) | 0.75 (0.69–0.81) |

ICC = intraclass correlation coefficient; CI = confidence interval; SD = standard deviation; EHR = electronic health record

mortality index (ICC 0.76). The absolute agreement ICCs was observed to drop to 0.62 in assessment of Elixhauser domain count, but no significant change in the absolute ICC was found for the Elixhauser index on readmission (ICC 0.79) and in-hospital mortality (ICC 0.75). Discrepancies between absolute and consistency ICC suggest the presence of non-negligible bias [33]. The occurrence of bias is further supported by the Bland-Altman analysis on Elixhauser domain count which demonstrated a mean difference of 1.4 domains (95% limits of agreement: -1.4 to 4.1) between chart review and EHR-based problem list measurement. The Bland-Altman analysis further demonstrated that only 6 (3.0%) subjects were outside the 95% limits of agreement (S2 Fig), which is within the expected 5% outside the limit of agreement. Lastly, ROC curve analyses yielded AUCs of 0.89, 0.86, and 0.83 in classifying the Elixhauser domain count, readmission index, and mortality index (S3 Fig).

## Discussion

This study shows that multimorbidity can be measured by applying the Elixhauser classification system to ICD-10-CM diagnostic codes stored in EHR-based problem lists. The EHR-based problem list performed with moderate-to-good accuracy with positive predictive value and negative predictive value of using the EHR-based problem list greater than 90% in 24/37 and 32/37 disease domains respectively. However, there was a clear bias towards undermeasurement as the problem list failed to identify 1.7 domains per patient. This result was largely driven by poor capture of two high prevalence disease domains including: obesity (NPV 65.3%) and uncomplicated hypertension (NPV 33.3). Despite these issues, the proposed methodology performed well overall as supported by high ICCs, Bland-Altman analysis (which only found 3% of subjects outside the 95% limits of agreement), and high AUCs observed in the ROC curve analyses.

Our findings have important implications for both practicing ED physicians and researchers. Use of the EHR-based problem list is particularly advantageous to ED physicians, whom have little familiarity with the patients they treat and often must make time-sensitive treatment decisions. Moreover, many ED patients (such as those with dementia) cannot self-report a complete medical history; a situation whereby ED physicians are nearly fully dependent on data in the EHR-problem list. The high specificities reported in this study demonstrate that reported diseases on the problem list generally represent true positives and may not require further verification through extended chart review. Regardless, ED physicians should be aware of underreporting of comorbid conditions and incorporate alternative methods—such as patient interview, medication reconciliation—to obtain a complete medical history. Secondly, physicians should be aware of specific patterns in underreporting of high prevalence conditions (i.e., obesity, depression, and hypertension) and seek to augment their documentation on EHR-based problem lists. Similar patterns in poor EHR capture of obesity and hypertension have been described elsewhere in the medical literature [17, 37], though the underlining cause remains unknown.

With regards to the research community, our findings support the use of ICD-10-CM diagnostic codes from EHR-based problem lists to serve as a foundation in describing multimorbidity. The proposed technique offers the inherent advantage of measuring multimorbidity at the point of care prior to any interventions being done in the ED. From a quality measurement perspective (i.e., measuring adherence to evidence-based guidelines), the method clearly delineates multimorbidity data that was available to physicians and advanced practice providers prior to performing any interventions. Furthermore, the described methodology—given its reliance on discrete, EHR-based variables—can readily be incorporated into more complex EHR-based implementation interventions, such as best practice alerts or embedded risk

stratification calculators. Such interventions have been shown to augment clinical workflow and improve guideline adherence [38]. Regardless, researchers should be aware of limitations in poor capture of obesity and hypertension. Augmented EHR-phenotypes incorporating both ICD-10-CM codes from the problem list as well extraction of additional discreet data elements —such as documented height & weight, laboratory results, radiographic findings, or ambulatory medications—could be incorporated into algorithms to improve the sensitivity and specificity in detecting the disease domains. Additionally, future algorithms to describe multimorbidity may incorporate advanced machine-learning methods, which can account for both structured data elements in the problem list as well as unstructured data captured in clinical notes.

## Limitations

First, we validated use of the EHR-based problem list in a narrow study population, namely older adult patients with HF at a single academic institution. This was consistent with our study hypothesis that patients with HF would have routine healthcare encounters with outpatient physicians, and thus a well-maintained problem list. Regardless, our findings will have to be validated in a broader population of ED patients and various types of hospitals. Secondly, we observed interrater variability in our gold standard measurement with moderate and good agreement in 5 and 7 domains, respectively. The gold standard relied on clinician raters as they can perform scoping chart reviews, though variability among clinicians is common and difficult to remove in highly controlled settings such as randomized trials [39]. Despite these challenges, our raters had near perfect agreement in the majority (18/37) of the domains. Thirdly, the AHRQ Elixhauser is missing some notable disease domains, specifically coronary artery disease and hyperlipidemia, that are relatively important in assessment of multimorbidity among older patients with HF. We believe these domains would be excellent additions to future iterations of the AHRQ comorbidity software. Finally, gold standard measurement of chronic conditions did not rely upon stringent guideline-based disease definitions (e.g., pulmonary function testing to confirm obstructive pulmonary disease). The average chart review was completed in an hour, thus use of stringent definitions for all the 37 disease domains examined was not feasible in this retrospective study. Further, we used information that is readily available to ED providers during health encounters. In this regard, the gold standard implemented in this study pragmatically reflects how physicians accrue multimorbidity data in clinical practice.

## Conclusion

Multimorbidity measurement from EHR-based problem lists can be performed with moderate-to-good accuracy among patients with HF in the ED. The methods do have some limitations with clear underreporting of two high prevalent disease domains (uncomplicated hypertension and obesity). Further work is needed to validate these findings in a broader array of ED patients and various types of hospitals.

## Supporting information

**S1 Table. Comparison of demographics of full population of ED patients with HF population as compared to selected random sample.** HF = Heart Failure.
(PDF)

**S2 Table. Reliability in measuring multimorbidity by structured chart review.**
*Indeterminate kappa due to complete agreement on either positive cases or negative cases

resulting in a 0 in the denominator; PABAK = prevalence-adjusted, bias-adjusted kappa.
(PDF)

**S3 Table. Diagnostic accuracy of measuring Elixhauser comorbid domains from the EHR-based problem list for all 37 domains.** EHR = electronic health record; PPV = positive predictive value; NPV = negative predictive value.
(PDF)

**S1 Fig. Flow diagram of selection of patients for the study cohort.**
(PDF)

**S2 Fig. Bland-Altman plots comparing Elixhauser domain count across chart review and EHR-based problem list.**
(PDF)

**S3 Fig. ROC analysis.**
(PDF)

**S1 Data.**
(XLS)

## Acknowledgments

We appreciate Hongtu Zhu and Huaying Qiu for providing feedback on the statistical approach.

## Author Contributions

**Conceptualization:** Brandon L. King, Michelle L. Meyer, Jan Busby-Whitehead, Martin F. Casey.

**Data curation:** Srihari V. Chari, Thomas Bohrmann, Martin F. Casey.

**Formal analysis:** Srihari V. Chari, Thomas Bohrmann, Martin F. Casey.

**Funding acquisition:** Brandon L. King.

**Investigation:** Brandon L. King, Karen Hurka-Richardson, Martin F. Casey.

**Methodology:** Brandon L. King, Michelle L. Meyer, Patricia P. Chang, Martin F. Casey.

**Project administration:** Michelle L. Meyer.

**Resources:** Brandon L. King, Srihari V. Chari, Thomas Bohrmann, Martin F. Casey.

**Software:** Srihari V. Chari, Thomas Bohrmann, Martin F. Casey.

**Supervision:** Patricia P. Chang, Martin F. Casey.

**Validation:** Michelle L. Meyer, Karen Hurka-Richardson, Thomas Bohrmann, Patricia P. Chang, Jo Ellen Rodgers, Martin F. Casey.

**Visualization:** Michelle L. Meyer.

**Writing – original draft:** Brandon L. King, Martin F. Casey.

**Writing – review & editing:** Brandon L. King, Michelle L. Meyer, Srihari V. Chari, Karen Hurka-Richardson, Thomas Bohrmann, Patricia P. Chang, Jo Ellen Rodgers, Jan Busby-Whitehead, Martin F. Casey.

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
