## [Decision Letter · Decision Letter 0]

13 May 2022

PONE-D-22-09414Accuracy of the Electronic Health Record’s Problem List in Describing Multimorbidity in Patients with Heart Failure in the Emergency DepartmentPLOS ONE

Dear Dr. Casey,

Thank you for submitting your manuscript to PLOS ONE. After careful consideration, we feel that it has merit but does not fully meet PLOS ONE’s publication criteria as it currently stands. Therefore, we invite you to submit a revised version of the manuscript that addresses the points raised during the review process.

We look forward to receiving your revised manuscript.

Kind regards,

Fabrizio Pecoraro

Academic Editor

PLOS ONE

Journal Requirements:

Reviewers' comments:

Reviewer's Responses to Questions

**Comments to the Author**

1. Is the manuscript technically sound, and do the data support the conclusions?

Reviewer #1: Partly

Reviewer #2: Yes

2. Has the statistical analysis been performed appropriately and rigorously? 

Reviewer #1: Yes

Reviewer #2: Yes

3. Have the authors made all data underlying the findings in their manuscript fully available?

Reviewer #1: Yes

Reviewer #2: Yes

4. Is the manuscript presented in an intelligible fashion and written in standard English?

Reviewer #1: Yes

Reviewer #2: Yes

5. Review Comments to the Author

Reviewer #1: Casey et al. present a study to evaluate a multimorbidity ‘problem list’ via electronic health records (EHRs) for patients with heart failure (HF) admitted in emergency departments (ED). The population sample was drawn from patients admitted in the ED of a southeastern US academic medical center. The inclusion criteria were determined on the basis of ICD-10 and demographics. An problem list across multiple disease domains was defined and multimorbidity was measured as gold standard via a 4-steps thorough chart review (plus additional materials if needed, e.g., echocardiography reports) performed by two trained persons and further checked by MD and nurse practitioner. The AHRQ Elixhauser (ICD-10) comorbidity score was calculated by manually reviewing ICD-10 codes. Domain based, and overall concordance indices for the cart review were calculated (i.e., PABAK, ICC), and then PPV and NPV of the gold standard with Elixhauser scores (three summary indices).

Results show a large range of variability in both the concordance indices and the diagnostic accuracy when stratifying by domain. In particular, there was poor capture of obesity and hypertension. The authors conclude that the diagnostic accuracy of the EHR-based multimorbidity problem list for patients with HF in ED.

Overall, the manuscript is well-written and well-structured, with balance among the presented sections.

However, there are concerns about the study rationale/premise, and robustness of analysis for which the conclusive claims or the clinical relevance might be hampered, diminished.

Methodological issues:

1. Study sample. First, ED/HF patients would possibly present as a selected population with comorbidity bias in one or more domains; the authors acknowledge that in part, but it may be a major driver. Second, the random sample was 200 over 1,130 eligible patients in the EHR data extraction. Even if chart review might have been too cumbersome for the whole sample, at least comparison with demographics and other descriptive statistics should have been made.

2. Most of EHR systems nowadays provide automated calculation of comorbidity index, usually the Charlson’s. The way in which the calculation is performed can vary, e.g., the time of lookback for finding a chronic condition, or the number of times a condition is diagnosed to confirm, and can be associated to a measurement error, population variance. The Elixhauser calculation presented here might have affected in the same way. The authors should have tried to consider a fully automated score provided by the EHR system.

3. The moderate PABAK and ICC in certain domains also indicates is substantial variability in clinical determinations, and this happens in parallel with the Elixhauser score determination, likely affecting the resulting diagnostic accuracy.

4. Given the continuous scales, a threshold analysis could have been useful (e.g., ROCs).

Clinical significance issues:

The work --as is-- stands as a mere exercise to compare imperfect, resource-intensive measures. The authors acknowledge the limitations. Yet, possibly, a predictive-based development based on these preliminary results could provide the translational relevance that does not emerge here.

Reviewer #2: Thank you for a neat study. This would be improved by including:

1. The means of random chart selection.

2. Assessment of key demographics (which ought to be readily suitable for automated statistical count) between the sample and the full non-excluded population - such as age, gender, ethnicity. This would confirm that the sample was representative in practice.

3. Discussion of the hypothetical savings which could be obtained in ED time if applied as a routine practice - e.g. saved time per patient; patients per day.

6. PLOS authors have the option to publish the peer review history of their article (what does this mean?). If published, this will include your full peer review and any attached files.

Reviewer #1: No

Reviewer #2: **Yes: **Professor Emeritus Michael Rigby

---

## [Author Response · Author response to Decision Letter 0]

5 Sep 2022

Reviewer #1: Casey et al. present a study to evaluate a multimorbidity ‘problem list’ via electronic health 

records (EHRs) for patients with heart failure (HF) admitted in emergency departments (ED). The 

population sample was drawn from patients admitted in the ED of a southeastern US academic medical 

center. The inclusion criteria were determined on the basis of ICD-10 and demographics. An problem list 

across multiple disease domains was defined and multimorbidity was measured as gold standard via a 4-

steps thorough chart review (plus additional materials if needed, e.g., echocardiography reports) 

performed by two trained persons and further checked by MD and nurse practitioner. The AHRQ 

Elixhauser (ICD-10) comorbidity score was calculated by manually reviewing ICD-10 codes. Domain 

based, and overall concordance indices for the cart review were calculated (i.e., PABAK, ICC), and then 

PPV and NPV of the gold standard with Elixhauser scores (three summary indices).

Results show a large range of variability in both the concordance indices and the diagnostic accuracy 

when stratifying by domain. In particular, there was poor capture of obesity and hypertension. The 

authors conclude that the diagnostic accuracy of the EHR-based multimorbidity problem list for patients 

with HF in ED.

Overall, the manuscript is well-written and well-structured, with balance among the presented sections.

However, there are concerns about the study rationale/premise, and robustness of analysis for which 

the conclusive claims or the clinical relevance might be hampered, diminished.

AUTHORS Response: Thanks for the positive feedback on our writing. We recognize the limitations of the 

proposed methodology, specifically that 1.7 domains were not identified per patient when using the 

EHR-based problem list. Although the proposed methodology for measuring multimorbidity has clear 

limitations, the method does form the basis of a promising foundation which can be built upon by 

extracting additional data elements (e.g., lab values, radiographic findings) to improve overall 

accuracy. We have updated the manuscript text to acknowledge this feedback (see page 9-10), which 

we very much agree with. 

Methodological issues:

1. Study sample. First, ED/HF patients would possibly present as a selected population with comorbidity 

bias in one or more domains; the authors acknowledge that in part, but it may be a major driver. 

Second, the random sample was 200 over 1,130 eligible patients in the EHR data extraction. Even if 

chart review might have been too cumbersome for the whole sample, at least comparison with 

demographics and other descriptive statistics should have been made.

AUTHORS Response: We have included a new supplemental table that compares the demographics of the 1,130 

eligible patients to the 200 patients randomly selected for chart review. We believe the values are 

comparable across the two groups (reflecting that random selection of charts for review was 

performed correctly; see Supplemental Table 1). 

2. Most of EHR systems nowadays provide automated calculation of comorbidity index, usually the 

Charlson’s. The way in which the calculation is performed can vary, e.g., the time of lookback for finding 

a chronic condition, or the number of times a condition is diagnosed to confirm, and can be associated 

to a measurement error, population variance. The Elixhauser calculation presented here might have 

affected in the same way. The authors should have tried to consider a fully automated score provided by 

the EHR system.

AUTHORS Response: We have investigated whether our healthcare system’s EHR includes any measures of 

comorbidity such the Elixhauser or Charlson index. Unfortunately, our current EHR build does not 

include any measures of comorbidity. Thank you for this suggestion. 

3. The moderate PABAK and ICC in certain domains also indicates is substantial variability in clinical 

determinations, and this happens in parallel with the Elixhauser score determination, likely affecting the 

resulting diagnostic accuracy.

AUTHORS Response: We included a discussion on this topic in our limitation section (see page 10). Thank you for 

this suggestion. 

4. Given the continuous scales, a threshold analysis could have been useful (e.g., ROCs).

AUTHORS: We maximized study power by using continuous scales, rather than thresholds. 

Unfortunately, the idea of multimorbidity measurement as a standalone measurement is still 

relatively nascent. We are unaware of any consensus agreements on meaningful thresholds that could 

be used to guide an ROC analysis comparing domain count and Elixhauser index scores. We do 

appreciate this suggestion, but do not see viable approach to implementing a meaningful ROC 

analysis. 

Reviewer #2

Clinical significance issues:

The work --as is-- stands as a mere exercise to compare imperfect, resource-intensive measures. The 

authors acknowledge the limitations. Yet, possibly, a predictive-based development based on these 

preliminary results could provide the translational relevance that does not emerge here.

AUTHORS Response: See above. We have expanded some language in our manuscript to highlight use of 

additional structured data elements (such as, laboratory values, radiographic findings) and advanced 

methods (such as machine-line) which incorporate unstructured data in capturing the burden of 

multimorbidity (see page 10).

Reviewer #2: Thank you for a neat study. This would be improved by including:

1. The means of random chart selection.

AUTHORS Response: We have a included a description and citation on the random sequence generator used to 

select charts in our methods. Thank you for this suggestion. 

2. Assessment of key demographics (which ought to be readily suitable for automated statistical count) 

between the sample and the full non-excluded population - such as age, gender, ethnicity. This would 

confirm that the sample was representative in practice.

AUTHORS Response: See above commentary on inclusion of new Supplemental Table 1. 

3. Discussion of the hypothetical savings which could be obtained in ED time if applied as a routine 

practice - e.g. saved time per patient; patients per day.

AUTHORS Response: We have expanded on the implications for ED physicians (see page 9), though with the 

caveats that the EHR-problem list is not a perfect technique. We are not aware of any studies on the 

exact time ED physicians spend obtaining medical history per patient, and thus are unable to quantify 

a meaningful hypothetical savings. We do appreciate this suggestion. 

Journal Requirements:

1. Please ensure that your manuscript meets PLOS ONE's style requirements, including those for file 

naming. The PLOS ONE style templates can be found at 

https://journals.plos.org/plosone/s/file?id=ba62/PLOSOne_formatting_sample_title_authors_affiliation

s.pdf

AUTHORS Response: We have reviewed the guidelines and updated the body of text accordingly. 

2. Please provide additional details regarding participant consent. In the ethics statement in the 

Methods and online submission information, please ensure that you have specified (1) whether consent 

was informed and (2) what type you obtained (for instance, written or verbal, and if verbal, how it was 

documented and witnessed). If your study included minors, state whether you obtained consent from 

parents or guardians. If the need for consent was waived by the ethics committee, please include this 

information.

If you are reporting a retrospective study of medical records or archived samples, please ensure that you 

have discussed whether all data were fully anonymized before you accessed them and/or whether the 

IRB or ethics committee waived the requirement for informed consent. If patients provided informed 

written consent to have data from their medical records used in research, please include this 

information.

AUTHORS Response: We have updated this section appropriately. 

3. Please update your submission to use the PLOS LaTeX template. The template and more information 

on our requirements for LaTeX submissions can be found at http://journals.plos.org/plosone/s/latex.

AUTHORS Response: We have reviewed the guidelines and updated the body of text accordingly. 

4. We note that the grant information you provided in the ‘Funding Information’ and ‘Financial 

Disclosure’ sections do not match. 

When you resubmit, please ensure that you provide the correct grant numbers for the awards you 

received for your study in the ‘Funding Information’ section.

AUTHORS Response: We have updated this section appropriately. 

5. In your Data Availability statement, you have not specified where the minimal data set underlying the 

results described in your manuscript can be found. PLOS defines a study's minimal data set as the 

underlying data used to reach the conclusions drawn in the manuscript and any additional data required 

to replicate the reported study findings in their entirety. All PLOS journals require that the minimal data 

set be made fully available. For more information about our data policy, please see 

http://journals.plos.org/plosone/s/data-availability.

Upon re-submitting your revised manuscript, please upload your study’s minimal underlying data set as 

either Supporting Information files or to a stable, public repository and include the relevant URLs, DOIs, 

or accession numbers within your revised cover letter. For a list of acceptable repositories, please see 

http://journals.plos.org/plosone/s/data-availability#loc-recommended-repositories. Any potentially 

identifying patient information must be fully anonymized.

AUTHORS Response: We have created a deidentified dataset to be uploaded that will be uploaded to our 

supplement in accord with PlosOne’s policies. Some variables such as gender and race were removed 

due to internal policies to ensure protection of private health information. These variables can be 

made available upon author request.

---

## [Decision Letter · Decision Letter 1]

28 Sep 2022

PONE-D-22-09414R1Accuracy of the Electronic Health Record’s problem list in describing multimorbidity in patients with heart failure in the emergency departmentPLOS ONE

Dear Dr. Casey,

Thank you for submitting your manuscript to PLOS ONE. After careful consideration, we feel that it has merit but does not fully meet PLOS ONE’s publication criteria as it currently stands. Therefore, we invite you to submit a revised version of the manuscript that addresses the points raised during the review process. As reported by Reviewer 1, some additional efforts should be made to investigate some of the issues previously highlighted by him/her, in particular with the domain determination (e.g., with a design change to assess robustness) and with the diagnostic accuracy. I suggest to update the paper on the basis of the Reviewer 1's comments before resubmit it to PLOS ONE. 

We look forward to receiving your revised manuscript.

Kind regards,

Fabrizio Pecoraro

Academic Editor

PLOS ONE

Journal Requirements:

Reviewers' comments:

Reviewer's Responses to Questions

**Comments to the Author**

1. If the authors have adequately addressed your comments raised in a previous round of review and you feel that this manuscript is now acceptable for publication, you may indicate that here to bypass the “Comments to the Author” section, enter your conflict of interest statement in the “Confidential to Editor” section, and submit your "Accept" recommendation.

Reviewer #1: (No Response)

Reviewer #2: All comments have been addressed

2. Is the manuscript technically sound, and do the data support the conclusions?

Reviewer #1: Partly

Reviewer #2: Yes

3. Has the statistical analysis been performed appropriately and rigorously? 

Reviewer #1: Yes

Reviewer #2: I Don't Know

4. Have the authors made all data underlying the findings in their manuscript fully available?

Reviewer #1: Yes

Reviewer #2: Yes

5. Is the manuscript presented in an intelligible fashion and written in standard English?

Reviewer #1: Yes

Reviewer #2: Yes

6. Review Comments to the Author

Reviewer #1: The authors have provided reasonable responses to my comments. However, rather than addressing the limitations of their paper, they have merely acknowledged them. I think that some additional effort should be made to investigate some of the issues previously highlighted, in particular with the domain determination (e.g., with a design change to assess robustness) and with the diagnostic accuracy. Computational phenotyping is an iterative process, but at the moment the paper still falls shorts of its premise and objective. Nonetheless, if the authors elaborate more on the domain and diagnostic performance, the manuscript could be published.

Reviewer #2: Thank you for addressing previous reviewer feedback, and providing a suitably improved and clearer manuscript.

7. PLOS authors have the option to publish the peer review history of their article (what does this mean?). If published, this will include your full peer review and any attached files.

Reviewer #1: No

Reviewer #2: No

---

## [Author Response · Author response to Decision Letter 1]

2 Nov 2022

As reported by Reviewer 1, some additional efforts should be made to investigate some of the issues 

previously highlighted by him/her, in particular with the domain determination (e.g., with a design change 

to assess robustness) and with the diagnostic accuracy. I suggest to update the paper on the basis of the 

Reviewer 1's comments before resubmit it to PLOS ONE. 

AUTHORS: We have incorporated a ROC curve analyses into our manuscript (pg 6, 9). Thank you 

for this suggestion as we believe it has made our manuscript more complete.

---

## [Decision Letter · Decision Letter 2]

29 Nov 2022

Accuracy of the Electronic Health Record’s problem list in describing multimorbidity in patients with heart failure in the emergency department

PONE-D-22-09414R2

Dear Dr. Casey,

We’re pleased to inform you that your manuscript has been judged scientifically suitable for publication and will be formally accepted for publication once it meets all outstanding technical requirements.

Kind regards,

Fabrizio Pecoraro

Academic Editor

PLOS ONE

Additional Editor Comments (optional):

Reviewers' comments:

Reviewer's Responses to Questions

**Comments to the Author**

1. If the authors have adequately addressed your comments raised in a previous round of review and you feel that this manuscript is now acceptable for publication, you may indicate that here to bypass the “Comments to the Author” section, enter your conflict of interest statement in the “Confidential to Editor” section, and submit your "Accept" recommendation.

Reviewer #1: All comments have been addressed

Reviewer #2: All comments have been addressed

2. Is the manuscript technically sound, and do the data support the conclusions?

Reviewer #1: Yes

Reviewer #2: Yes

3. Has the statistical analysis been performed appropriately and rigorously? 

Reviewer #1: Yes

Reviewer #2: Yes

4. Have the authors made all data underlying the findings in their manuscript fully available?

Reviewer #1: Yes

Reviewer #2: Yes

5. Is the manuscript presented in an intelligible fashion and written in standard English?

Reviewer #1: Yes

Reviewer #2: Yes

6. Review Comments to the Author

Reviewer #1: No additional comments. No additional comments. No additional comments. No additional comments. No additional comments.

Reviewer #2: The additional material enhances an already acceptable paper. Thank you for your careful considerations.

7. PLOS authors have the option to publish the peer review history of their article (what does this mean?). If published, this will include your full peer review and any attached files.

Reviewer #1: No

Reviewer #2: No

---

## [Editor Report · Acceptance letter]

2 Dec 2022

PONE-D-22-09414R2 

Accuracy of the electronic health record’s problem list in describing multimorbidity in
patients with heart failure in the emergency department 

Dear Dr. Casey:

I'm pleased to inform you that your manuscript has been deemed suitable for publication in PLOS ONE. Congratulations! Your manuscript is now with our production department. 

Kind regards, 

on behalf of

Dr. Fabrizio Pecoraro 

Academic Editor

PLOS ONE